# Deep ConvNet: Non-Random Weight Initialization for Repeatable Determinism, Examined with FSGM [note 1]

**DOI:** 10.3390/s21144772

**Published:** 2021-07-13

**Authors:** Richard N. M. Rudd-Orthner, Lyudmila Mihaylova

**Affiliations:** Department of Automatic Control and Systems Engineering, University of Sheffield, Sheffield S1 3JD, UK; L.S.Mihaylova@sheffield.ac.uk

**Keywords:** repeatable determinism, weight initialization, convolutional layers, adversarial perturbation attack, FSGM, transferred learning, machine learning, smart sensors

## Abstract

A repeatable and deterministic non-random weight initialization method in convolutional layers of neural networks examined with the Fast Gradient Sign Method (FSGM). Using the FSGM approach as a technique to measure the initialization effect with controlled distortions in transferred learning, varying the dataset numerical similarity. The focus is on convolutional layers with induced earlier learning through the use of striped forms for image classification. Which provided a higher performing accuracy in the first epoch, with improvements of between 3–5% in a well known benchmark model, and also ~10% in a color image dataset (MTARSI2), using a dissimilar model architecture. The proposed method is robust to limit optimization approaches like Glorot/Xavier and He initialization. Arguably the approach is within a new category of weight initialization methods, as a number sequence substitution of random numbers, without a tether to the dataset. When examined under the FGSM approach with transferred learning, the proposed method when used with higher distortions (numerically dissimilar datasets), is less compromised against the original cross-validation dataset, at ~31% accuracy instead of ~9%. This is an indication of higher retention of the original fitting in transferred learning.

## 1. Introduction

Convolutional layers in neural networks have been used in Artificial Intelligence (AI) applications and led to the use of multiple layers separated by non-linearity functions. This layering of hidden layers is said to be deep, and the successes of that architecture led to the Deep Learning (DL) research thread. It is generally accepted that convolutional layers may have some translation to brain anatomy with respect to Hubel and Wiesel [1,2]. Who examined spider monkeys’ and cats’ brain activity when under a light anesthetic while stimulating the retina with images of spots, stripes and patterns. Thus convolutional layers have had biological inspirations, and are generally accepted as providing hierarchical feature extraction in a Deep Convolutional Network (ConvNet) [3,4]. Later in 2012, Alex Krizhevsky’s paper [5] would prove to become an influential paper, and demonstrated better than human performance within image categorization in the image net challenge using deep convolutional networks. Convolutional Neural Networks (CNN) have played an influential role ever since. Although applications of convolutional layers provide some important human level capabilities, they have not been embraced into mission critical applications [6,7,8,9], owing in part to learning session accuracy variations, and certification of the network content as complete and correct. The current initialization methods using random numbers provide a cross-validation variation in accuracy that is visible over regularization, that is to say, that different random initialization states provide variations in the prediction accuracy when cross-validated. This research focuses on the learning session variations for repeatable determinism via the use of non-random initialization states. That is a desirable quality for a safety critical application, and is in tune with raising capability closer to human decision levels in safety applications. Although this paper is disregarding any ethical or legal liability issues of a computer making decisions. Convolutional networks have still been proposed in applications where a lower service level can be tolerated, a liability can be avoided by making them limited to advisory support, or where financial penalties can be understood estimated and underwritten. Applications include: river side monitoring to avoid port saturation [10], traffic accident detection and condition analysis based on social networking data [11]. There is also a recent appetite in research for higher safety certification in the application of driverless cars [12] by Holen et al. in 2020, and by Fremont et al. in the use of specialist toolkits like VerifAI, used in an AI controlled aircraft taxi system, and applied in the development methods for detecting, isolating and correcting errors by deducing error conditions [13]. Furthermore, Thombre et al. in 2020 [14], considers Gaussian processes and Machine Learning (ML) in a review of AI approaches, for multi-sensor data-fusion, and sensor integrity in the application of future autonomous ships. The approach in this paper is in support of higher integrity data assurance of AI, but more specifically convolutional networks in image processing.

The previously published background work [15] to this paper, also examined repeatable determinism, but in perceptron layers only, and proposed a method, although was tightly coupled to the perceptron layers. That previous paper resolved a numerical stability issue for repeatability and proposed a non-random method for determinism, which achieved an almost equal performance, and it proved the viability of a non-random method without sampling the dataset. That work was furthered in a journal published version [16], which used the Glorot/Xavier limit values, and then achieved an equal performance in accuracy to the random method, now proving equality in performance as an alternative approach. See Figure 1 (left) for the weight matrix as an image of learned weights with the existing random method, and Figure 1 (right) reproduced from the previous work’s [16] non-random number method, both in perceptron layers.

Noted in that previous work [16], that both weight sequences had an equivalence in performance, but the non-random method (in Figure 1 (right)) has a structure that may have a benefit for rule-extraction. As the weights have been ordered into a sequence along the number of neurons (in the neurons axis) and show activation correlations at pixel positions (in the pixel activations axis), and that helps to generalize in a rule extraction approach, as the pixel activations have been clustered to neighboring weights.

However, those previous papers [15,16] were confined to perceptron layers and this paper furthers that work into convolutional networks. That earlier work [15,16] in perceptron layers did prove that an equal performance of a non-random weight initialization method was viable and that random numbers for the initialization are not necessary. This is the same assertion of Blumenfeld et al. too in 2020 [17], in an experiment of zeroing of some of the weights in a convolutional layer. However, the zeroing of weights is not the approach presented here. Furthermore, the order of weights was not significant in the background perceptron work [15,16], due to the fully connected links of nodes, and in convolutional layers, the weights relate to convolved spatial filters, and so the order is significant. So the previous perceptron form [15,16] is not directly applicable within convolutional layer networks.

The paper’s structure is as follows: Section 1 is the introduction to the area with the background work, recent contemporary related works, and the novelty of this method. Section 2 is the experiment benchmark baseline model. Section 3 presents the proposed non-random method and explains the design. Section 4 compares the existing random initialization method with the proposed method. It also verifies robustness to He et al. initialization limits as the current state-of-the-art. Section 5 makes a comparison under adversarial perturbation attack using the Fast Sign Gradient Method (FSGM) with transferred learning, and applies the proposed method to a color image dataset (MTARSI2), in a dissimilar architecture. Section 6 presents the discussion of results and concludes the paper.

### 1.1. Related Work

Seuret et al. [18] in 2017, proposed using Principal Component Analysis (PCA) to initialize neural layers by generating an autoencoder for the PCA parameters in the application of document analysis, and in that application outperformed random methods. Seyfioğlu et al. [19], in 2017, applied transferred learning and unsupervised convolutional autoencoder (CAE) in the application of radar microDoppler where only small datasets are available. Seyfioğlu et al. found that transferred learning was superior on small datasets (less than 650 samples), and CAE in larger datasets. However, both approaches outperformed the random method in both dataset sizes. Humbird at al. [20], in 2019 proposed a Deep Jointly Informed Neural Network (DJINN) for neural network initialization using decision trees. This approach still employs random numbers in the weights, and in the sampling of a normal distribution in the biases. It is searching for the warm start condition in learning with a dataset used in back-propagation. Zhang et al. [21] in 2019 highlight the area of initialization as an active research topic, and propose FIXUP Initialization as a modification to other initialization methods limit values. Zhang et al. claim with the proper regularization, FIXUP allows 10,000 layers without normalization. Ding et al. in 2020 [22], proposed a shuffle leapfrog algorithm approach, for the update and initialization with random Gaussian forms in the area of fundus lesions images. The approach presented in that paper contains random numbers, initially in a Gaussian distribution optimized with the shuffle leapfrog algorithm, whereas the approach presented in this paper, avoids the use of random numbers for repeatable determinism. Wang et al. in 2020 [23], proposed a 2D Principle Component Analysis (2DPCA) approach to the initialization of convolutional networks to adjust the weight difference values to promote back propagation. This approach avoids the use of random numbers and uses samples of the dataset instead, making it convergent to the sample data seen. Ferreira et al. in 2019 [24], examined weight initialization using a De-noising Auto-Encoder (DAE) in the field of classifying tumor samples through dataset sampling, a data sample convergent approach. In 2021, Lyu et al. [25], in neuroevolution point out three weight initialization points (creation, offspring and mutation). They use Evolutionary eXploration of Augmenting Memory Models (EXAMM) neuroevolution algorithm to generate Recurrent Neural Networks (RNN): ∆-RNN, Gated Recurrent Units (GRU), Long Short-Term Memory (LSTM), Minimal Gated Units (MGU) and Update Gate RNN (UGRNN). Lyu et al. assess during crossover and mutation operations: Xavier and Kaiming (also known as He), and two neuroevolution Lamarckian weight inheritance methods. They find that Lamarckian weight inheritance is superior in crossover and mutation operations, but the method still uses random numbers. In contrast, the method presented here is aligned to image classification, rather them neuroevolution, and is replacing random numbers rather than adapting them. In the proposed method the approach is not coupled to the sample data, only the architecture in terms of layers and is adaptive for filter geometries and layer types used.

### 1.2. Contribution and Novelty

This paper is an extended version of the paper presented at IEEE DeSE 2020 conference [26], and uses Tensorflow 2.0 instead in its’ results, and also with an extension of the FSGM approach. The related work approaches fall into four categories: (1) sampling, pre-processing datasets and weight selection, (2) optimizing or normalizing limit values to the activation functions used, (3) seeding, saving and retrieving or transferred learning, (4) optimizing the selection of random sequences. However, the proposed method approach presented is arguably a fifth category, and that is substituting the random numbers with a non-random sequence without data sampling that is predisposed to the application generically. This approach could indeed be also applied to the CNN approaches of image classification in the related work as the initial condition supporting them. The previous work [15,16], also arguably fell into this fifth category as an alternative weight initialization sequence without data sampling. Although the previous work’s [15,16] focus was for repeatable determinism rather than performance, to support dependable systems. It achieved an equal accuracy, but in this approach presented the accuracies are higher than the random methods, and it is also not a data sampling or pre-processing approach, thus making it not limited to that dataset. This paper’s contribution is a proposed alternative initialization method, for generating a non-random number sequence for the initialization of convolutional and perceptron layer mixes, rather than perceptron layer only networks as in the previous work [15,16]. The proposed non-random number sequence has formations of stripes and curves in that initialization state, and as such is predisposed to the application of image categorization, and it is independent of the dataset utilized. It also allows earlier learning, to lower the loss quicker, and arrives at a higher performing accuracy in the first learning session, that is repeatable and deterministic. That is supporting a value of future dependable systems in mission and safety critical applications of smart sensing. With the existing random number methods, several learning sessions are required to establish which learning session’s random sequence has provided the best accuracy from a variety of random initialization states. In comparison to the related work approaches, that required to: adjust weights, use random numbers or sample the dataset for convergence. The proposed method is without data sampling or random sequences as a more general case. This is also a complementary approach to normalization and limit optimization approaches like Glorot/Xavier, and He et al. initialization limit values [27]. The method proposed substitutes only the use of random numbers for a deterministic non-random finite number sequence and retains the number range limits of the original methods.

Although transferred learning is not part of the proposed method, the transferred learning approach is used as an analytical technique with the FSGM approach. This examines the performance with a controlled distortion between the model being transferred and a dataset in subsequent learning that contains the FSGM distortion. As such the model using the proposed method, after model defense with transferred learning with an FSGM dataset, has less compromised the original dataset training with larger epsilon (*ε*) value distortions of the FSGM attack. The intuition for this is that the existing random initialization method provides an unintentional noise source, which causes noise re-colorization when combined with the noise sources in the images delaying learning, while also being less predisposed to the application. The FSGM with transferred learning approach provides a convenient method, for examining the transferred learning compromise with a controlled distortion via the epsilon (*ε*) value. Although neither method is immune to FSGM attacks, which is not the objective of the technique in this analysis, the proposed method has the advantage of been less compromised to the original dataset, with transferred learning in model defense when the datasets are more divergent (larger distortions as a numerically dissimilar dataset).

Some distinctions of this work are:Arguably part of a new category of weight initialization type,Does not redefine limit values, as such is compatible to them,A replacement to random numbers, that is aligned to image classification,Is detached from the dataset used, without data sampling,Repeatable and deterministic that is aligned to safety critical applications,Higher retention of previous model content in transferred learning,Applicable to other methods in combination, (i.e., complementary).

## 2. Experiment Benchmark Baseline Model

As the previous work [15,16], perceptron layers and the MNIST dataset [28], were used as it is familiar to researchers. So to demonstrate non-random weight initializations the same application and dataset are used but in a convolutional form. This benchmark is also used such that comparisons can be made from the previous background work [15,16] as well. Figure 2 presents the architecture of the benchmark model in the convolutional layer form.

The model architecture is the equivalent of the perceptron layer foundation work’s benchmark [15,16], but in a convolutional layer form, and as such forms a comparison bridge to the background work [15,16]. Using the repeatable critical-sections defined from the background work [15], which removed a source of numerical instability in learning session variations. The convolutional layer benchmark results are presented using the Glorot/Xavier random number method initialization as per its definition by Torres [29] and has a stated accuracy of about ~97%. Although it should be noted, that there are higher scoring models using the MNIST dataset in a convolutional form, a high accuracy score of 99.8% by Kassem [30], provides little headroom to show an improvement. That model also requires 50 epochs, and that is a long learning duration beyond the initial condition in this context. Where the random shuffle may be a more dominant random effect. The results of the baseline are: an average accuracy of 89.575% in a single epoch with no shuffle, 91.128% when a shuffle is used, and 96.901% with the use of five shuffled epochs, reaching the stated accuracy of the Torres [29] model.

### Understanding the Weights and Image Sizes

To understand the presented proposed non-random weight method, the structure of the weights and the image sizes, in the benchmark model need to be understood clearly. To understand how the weights are used is critical, to understanding the receptive field in subsequent layers. As this is quite different from perceptron layers, and as in the findings of the journal version of the perceptron repeatable determinism paper [16]. When using the Glorot/Xavier limits [31], the results were enhanced with the non-random method [16], over the results initially presented in the conference paper [15]. This was because of the tolerance and matching to the model architecture in terms of propagation updates and limits.

Convolutional layers use the weights for the filters, and not the pixels directly, as such the dimensions of each filter are: width by height, then by depth (channels), where that depth may be inherited from the previous convolutional layer’s filters. Perceptron layers in a ConvNet use the image size by the previous layer’s filters, as the previous layer’s filter would have translated to depth (channels) in activations, and those activations are connected to each neuron. Thus, the Gloror/Xavier limits need to be calculated, and Figure 3 illustrates the inheritance and hyper-parameters effect on the image and weights size parameters in the benchmark.

In Figure 3, on the left hand side of the Torres model, are the number of weights given the layer hyper-parameters. On the right hand side are the image size adjustments in the convolutional layer, which are a tensor of activations, presented to a flattening layer to become a vector for the final dense layer.

Extracted from Figure 3, Table 1 shows the weights and image sizes in each layer in the benchmark model. Such that the Glorot/Xavier limit values can be calculated to support the number ranges in the Glorot/Xavier initialization approach, and also later in the state-of-the-art He et al. initialization approach as well.

To calculate the Glorot/Xavier limits, see Equations (1)–(3), but note that the calculated values have been rounded to eight decimal places (as a rule of thumb for precision [32]), and are used as such and are shown as such in the Equations (1)–(3):(1)$$ConvLayer1=\sqrt{\frac{6}{\left(5\cdot 5\cdot 1+5\cdot 5\cdot 1\cdot 32\right)}}=0.08528029,$$
(2)$$ConvLayer2=\sqrt{\frac{6}{\left(5\cdot 5\cdot 1\cdot 32+5\cdot 5\cdot 64\right)}}=0.05\;\mathrm{and}$$
(3)$$DenseLayer=\sqrt{\frac{6}{\left(4\cdot 4\cdot 64+10\right)}}=0.07617551.$$

Alongside the Glorot/Xavier limits, the structure of the weight initialization sequence also requires to have positional stripes and curves variations in each filter. Those stripes and curves positional variations are also to be aligned to feature extraction, in a Hubel and Wiesel [1,2] stripes intuition. That structure is ideal to be more allied to edge detection than a random value placement as the start condition. This is to predispose the initial condition more generically to the application of image classification. Thus outperform the random methods, by inducing earlier learning, with less unlearning of the initial state, and using the early data in the dataset more effectively.

## 3. The Proposed (Non-Random) Method

To explain the design of the proposed method, and how it is derived. The intention was that the proposed initialization method would make filter arrangements that have stripes, spots and curves that are different in each filter. Such that the subsequent learning adapts to the dataset quicker being pre-disposed to the application. These arrangements are also different in each filter, providing a filter diversity of edge detection, in different orientations. As the positional variation of values is important as it relates to a filter sweeping across the pixels. If the modulation of arrangement position is based on a filter cell multiple (like two) in a matrix, then alternations may relate to stripes in a 2D matrix when different maximum width values are used, which would be controlled from a hyper-parameter for that layer: filter width. This would also connect the filter to the resolution in that filter in the model layer. That striping will then be controlled by the convolutional layer’s hyper-parameters (filter height, width and number of filters).

As also the stripe orientation in different filter arrangements is important, a diversity is required in each filter over the number of filters. An algorithm published in the paper [33], that produced a least adjacent arrangement based on a modulation of two for dataset shuffling, has some attractive properties to this application. It was originally an alternative to the established random dataset shuffle approach. This non-random shuffle approach, rearranges the dataset to produce a sequence with: the first half of the input that was output at a stride of two, and then infilled the gaps with the remaining vector, also at a stride of two but in reverse order. This resulted in a placement with the smallest and largest numbers neighboring each other at the start of the vector. This process can be repeated iteratively as an in place operation, and provides a number of filter variations that are deterministic.

When iteratively repeated as an in place operation, the original sequence order will repeat nominally, at a maximum number of iterations of: *vector length − 1* or less, and is an iterative numerical sequence. This algorithm is to be used as a readdressing method of the initialization weights, pertaining to the filters to provide stripes and curves in that initialization method. The algorithm from the paper [33], is further enhanced here to deal with odd number length vectors.

The output of the function *shuffle* is a set *y*, with a re-ordering of the set *x* as defined by the subscripts *α* and *β*. As in logically moving from address position *α* to position *β*. The *shuffle* function declared in Equation (4) is recursive, where the number of recursions is defined by the number of filters (*nFilter*) in that layer as a subscript (*_LayerNo_*). The formulas for the algorithm are in Equations (4)–(7) as amended and use zero indexing.
(4)$$y=shuffle\left(x,nFilter_{\left(LayerNo\right)}\right).$$

The *shuffle* function is defined in Equation (5) using equation guards, that if the *nFilter_(LayerNo)_* (or *i*) is greater than one filter, then the recursion is still made with the subscript reordering (*α*), while decrementing the *i* value by 1 on each recursion iteration. These recursions occur until the last recursion that will return the unordered subscript location (*β*). Which will then be subject to all the subscript reorder recursions as the function shuffling iterations prior are applied to complete the shuffle definition pattern:(5)$$shuffle\left(x_{\left\{\beta \right\}},i\right)=\left\{\begin{array}{cc} shuffle(x_{\left\{\alpha \right\},}i-1) & if\;i>1\\ x_{\left\{\beta \right\}} & otherwise \end{array}\right.,\;$$
where the index subscript set for ‘*β*’ is in Equation (6), and where *n* defines the set size (and again is zero indexed):(6)$$\beta_{\left\{0..n-1\right\}}=\left\{\beta \in N|\begin{array}{c} \left\{0\leq 2\beta \leq 2\left(\lfloor \frac{n}{2}\rfloor -1\right)\right\}\cup \\ \left\{1\leq \left(2\beta +1\right)\leq \left(2\left(\lfloor \frac{n}{2}\rfloor -1\right)+1\right)\right\}\cup \\ \left\{=\{\begin{array}{cc} \left(n-1\right) & if\;n\left(mod\;2\right)\neq 0\\ \left\{\right\} & otherwise \end{array}\}\right\} \end{array}\right\},$$
and where the last set union is included if the set length (*n*) is an odd number defined by *n (mod* 2) ≠ 0, as the modulo division of the set length (*n*) by modulus 2. This was the amendment from the paper [33]. The index subscript set for *‘α’* is in Equation (7), and naturally has the same set size (*n*) and forms the initial order for the re-ordering displacement subscripting in the shuffle pattern:(7)$$\alpha_{\left\{0..n-1\right\}}=\left\{\alpha \in N|\begin{array}{c} \left\{0\leq \alpha \leq \lfloor \frac{n}{2}\rfloor -1\right\}\cup \\ \left\{n-1\leq n-1-\alpha \geq \lceil \frac{n}{2}\rceil \right\}\cup \\ \left\{=\{\begin{array}{cc} \left\lfloor \frac{n}{2}\right\rfloor & if\;n\left(mod\;2\right)\neq 0\\ \left\{\right\} & otherwise \end{array}\}\right\} \end{array}\right\},$$

In illustration, *x* at this point can be thought of as a sequence of numbers as in Equation (8). However, the intended set values for *x* will be further defined later in this paper in the *valSet* function:(8)$$x_{\left\{0..n-1\right\}}=\left\{0..\left(n-1\right)\right\}\;\mathrm{and}:\;\mathrm{where}\;n\;\mathrm{is}\;\mathrm{the}\;\mathrm{length}\;\mathrm{of}\;\mathrm{the}\;\mathrm{tensor}.$$

This algorithm will always have the same value in the first location. Although this was not significant in the dataset shuffle application in the paper [33], it is significant in this application with spatial convolutional filters. Experiments were conducted with pre-placement shift offsets in the data, and also with a data direction variation in the *shuffle* algorithm. These experiments proved to not be as high performing, although did provide a higher number of unique filters. As in convolutional layers, the order of filter values is significant, so a pre-alternation of the data is conducted instead. So that every second filter is reversed (or flipped) and the memory is addressed through width, height and depth for the odd filter numbers, and vector address reversed as then depth, height, width in reverse order for the even filters. This provides two different filters of alternating direction in the same shuffle iteration, doubling the number of unique filters on offer, with a crucial disruption of the first position value as its primary intention.

As well as the address order placement, the value distribution of the values within the initialization sequence can be significant, as images are less likely to be uniformly distributed in pixel values. Experiments were conducted with linear ramps, as these had been the highest performing in perceptron layers [15,16]. In these experiment cases, the application of a linear ramp was higher performing with dense perceptron layers and sinusoidal slopes in the convolutional layers. This might be because the *cos*(*x*) content is a partial distribution of a *sine* function (bath-tub), or at least its distribution has a match to convolutional layers, and the image data it processes. As such the sinusoidal slope and linear ramp are selected based on the layer type within the network model architecture, because of the indirect image processing in the convolutional layers.

The formulas that call the addressing shuffle function (*shuffle*) are in Equations (9)–(30), and include the addressing alternation in the definition. Note that it also calls a function (called *valSet*) that provides a response based on the value ratio (*cnt/m*), and network layer type (*t*) to select a sinusoidal slope or linear ramp value form, altering the value distribution between layer types. The initialization tensor length (*InitTensor_Length_*) for a layer is based on a number of filters (*maxFilters*) of that layer and the number of weights in the filter (*m*) is as in Equation (9):(9)$$InitTensor_{Length}=m\cdot maxFilters.$$
where each convolutional layer’s filter tensor length can be calculated as (*m*), which is from the convolutional layer’s filter: height, width and depth as a 3D matrix size and is defined as in Equation (10). The value of *m* provides a maximum scale value (as the denominator of a ratio) for a numerator value *cnt* (as a progressive weight *count* in the filter), within each filter of a convolutional layer. Where *m* − 1 is the maximum value that the value that *cnt* as part of a ratio can achieve as defined in Equation (11).
(10)$$m=Height_{Filter}\cdot Width_{Filter}\cdot Depth_{Filter},$$
(11)$$cnt=\left\{0..m-1\right\},\;indexed\;as:\;cnt_{\left(Width_{Filter},\;Height_{Filter},\;\cdot \;Depth_{Filter}\right)\;},$$

Note that the weight calculation algorithm of the dense layer is dependent on the layers prior, and as such the dense layer weight calculation algorithm may vary depending on the architecture. This is because the activations could be the number of neurons of a proceeding dense layer, and in that case, the subsequent shuffle reordering and flips may not be necessary as the layers are fully connected as in the papers [15,16]. However, if there is a proceeding convolutional layer, then the activations map to the receptive fields of the convolved image filters, which is the case in this benchmark model. So in this case the height and width are the image size and the depth is inherited as the channel depth (or filters from the previous layer), as in Equation (12), and the value set of *cnt* is in Equation (13).
(12)$$m=Height_{Image}\cdot Width_{Image}\cdot Depth_{image},$$
(13)$$cnt=\left\{0..m-1\right\},\;indexed\;as:\;cnt_{\left(Width_{image},\;Height_{image},\;\cdot \;Depth_{image,}\right)}.$$
where *nFilter* (or *nNeurons* for a dense layer) and *nDepth* are number sets that are zero indexed, and the limit of *nFilter* is *maxFilters* − 1 (or *maxNeurons* − 1 for a dense layer), and *nDepth* is *maxDepth* − 1. Those sets are defined in Equations (14)–(16).
(14)nFilter=nFilter∈N |0≤nFilter<maxFilters, for a convolutional layer,
(15)nNeurons=nNeurons∈N |0≤nNeurons<maxNeurons, for dense layer, and
(16)nDepth=nDepth∈ℕ |0≤nDepth<maxDepth.

As such, each filter vector (or *neurons vector* in a dense layer) will be a vector of values with a *vector length* of *MaxFilters* (*or MaxNeurons*), and with repeating values in the set in Equation (17):(17)nSet=nSet∈ℕ |0≤cnt<m−1.

The initialization tensor is a 4D tensor matrix of a *nHeight* 3D matrix tensor that comprises a 2D matrix of *nWidth*, and that is a *nDepth* 1D vector of *nFilter* length as the subscripts illustrated in the Equation (18):(18)$$InitTensor={\;}_{\left(nHeight,\;nWidth,nDepth,nFilter\right)},\;\mathrm{for}\;a\;\mathrm{convolutional}\;\mathrm{layer}.$$

In this model test case, the dense perceptron layer’s initialization tensor is re-indexed from the receptive field mapping of the convolved filters in a previous layer to a matrix of *activations* and *neurons* as the subscripts in the Equation (19):(19)$$\;InitTensor_{\left(Activations,Neurons\right)}={\;}_{\left(\left(nHeight,\;nWidth,nDepth\right),nNeurons\right)},\;\mathrm{for}\;a\;\mathrm{dense}\;\mathrm{layer}.$$
where the set for the subscripts *nHeight* and *nWidth* are given as in Equations (20) and (21) and are the filter geometry in convolutional layers, or the image geometry mapped to the convolved filters in dense perceptron layers, when following a convolutional layer:(20)$$nHeight=\left\{nHeight\in N\;|0\leq nHeight<maxHeight\right\},$$
(21)$$nWidth=\left\{nWidth\in N\;|0\leq nWidth<maxWidth\right\}.$$

A convolutional layer illustrative example of the *cnt* values (convolved filter addressing) is given in Equation (22), in the case of 5 filters with a channel depth of 4 and the filter dimensions of width 3 and a height of 2.
(22)$$set_{of_{cnt}}=\left\{\begin{array}{c} \left\{\begin{array}{c} \left\{0,\;0,\;0,\;0,\;0\right\},\;\\ \left\{6,\;6,\;6,\;6,\;6\right\},\;\\ \left\{12,\;12,\;12,\;12,\;12\right\},\;\\ \left\{18,\;18,\;18,\;18,\;18\right\} \end{array}\right\},\;\\ \left\{\begin{array}{c} \left\{1,\;1,\;1,\;1,\;1\right\},\;\\ \;\left\{7,\;7,\;7,\;7,\;7\right\},\;\\ \left\{13,\;13,\;13,\;13,\;13\right\},\;\\ \left\{19,\;19,\;19,\;19,\;19\right\} \end{array}\right\},\;\\ \left\{\begin{array}{c} \left\{2,\;2,\;2,\;2,\;2\right\},\;\\ \left\{8,\;8,\;8,\;8,\;8\right\},\;\\ \left\{14,\;14,\;14,\;14,\;14\right\},\;\\ \left\{20,\;20,\;20,\;20,\;20\right\} \end{array}\right\} \end{array}\right\}\left\{\begin{array}{c} \left\{\begin{array}{c} \left\{3,\;3,\;3,\;3,\;3\right\},\;\\ \;\left\{9,\;9,\;9,\;9,\;9\right\},\;\\ \left\{15,\;15,\;15,\;15,\;15\right\},\;\\ \left\{21,\;21,\;21,\;21,\;21\right\} \end{array}\right\},\;\\ \left\{\begin{array}{c} \left\{4,\;4,\;4,\;4,\;4\right\},\;\\ \;\left\{10,\;10,\;10,\;10,\;10\right\},\;\\ \;\left\{16,\;16,\;16,\;16,\;16\right\},\;\\ \left\{22,\;22,\;22,\;22,\;22\right\} \end{array}\right\},\;\\ \;\left\{\begin{array}{c} \left\{5,\;5,\;5,\;5,\;5\right\},\;\\ \left\{11,\;11,\;11,\;11,\;11\right\},\;\\ \;\left\{17,\;17,\;17,\;17,\;17\right\},\;\\ \left\{23,\;23,\;23,\;23,\;23\right\} \end{array}\right\} \end{array}\right\}.\;\mathrm{for}\;a\;\mathrm{Conv}\;\mathrm{layer}$$

Equation (22) shows vectors of five filters are provided for each of the four depth channels, and those, in turn, are for each of the filter dimensions of width (3) and height (2). The values of *cnt* are counting through values of width then height and then depth as an indexing order and at this point, all filter weight values of *cnt* are matching, as latterly they will be shuffled and alternated towards the final filter permutations for linear striping. The tensor of values of *cnt* with respect to *m* are applied to the slope alternatives that are as in Equation (23), and that can select between the convolutional and perceptron layer type (*t*), also with the chosen calculated limit value as (*l*) for either uniform He et al. or Glorot/Xavier limit values.
(23)$$valSet\left(cnt,m,l,t\right)=\left\{\begin{array}{cc} cos\left(\frac{cnt}{m-1}\pi \right)l & if\;t=Convolutional\\ \frac{cnt}{m-1}2l-l\; & if\;t=Perceptron \end{array}\right..$$

The *valSet* function provides two value sequence types depending on the layer type, this provides two distributions of values that are either uniformly distributed for dense layers, or bathtub distributed in nature for the convolutional layers. As the shuffle reordering does not shift the address, the value of every second filter is reversed in order, for convenience this is done in a matrix transpose, as in Equation (24).
(24)$$TsprMat_{\left(nFilter,\;nDepth,\;nWidth,\;nHeight,\right)}=T_{\left(initvalues_{nHeight,\;nWidth,\;nDepth,\;nFilter}\right)}$$

Every second filter is order reversed (flipped) as a contiguous vector tensor as in Equation (25):(25)$$flipMat=\left\{\begin{array}{cc} TsprMat_{\left(nFilter,\left(m-1\right)..0\right)} & \left(nFilter+1\right)\;\left(mod\;2\right)=0\\ TsprMat_{\left(nFilter,0..\left(m-1\right)\right)} & otherwise \end{array}\right.\;$$

Then the vector is re-indexed back to the matrix subscripts as in Equation (26):(26)$$TsprMat2\left[nFilter\right]=flipMat_{\left(Depth,Width,Height\right)}\;$$

Again for illustration, if the matrix (*TsprMat2*) is also transposed back and using the *cnt* values, rather than the *valSet* function response values as intended, so as to provide a clear illustration in comparison with the previous example in Equation (22). Then the matrix becomes as in Equation (27) and shows the filter flips in every second filter when compared with Equation (22).
(27)$$set_{of_{cnt}}=\left\{\begin{array}{c} \left\{\begin{array}{c} \left\{0,\;23,\;0,\;23,\;0\right\},\;\\ \left\{6,\;17,\;6,\;17,\;6\right\},\;\\ \left\{12,\;11,\;12,\;11,\;12\right\},\;\\ \left\{18,\;5,\;18,\;5,\;18\right\} \end{array}\right\},\;\\ \left\{\begin{array}{c} \left\{1,\;22,\;1,\;22,\;1\right\},\;\\ \;\left\{7,\;16,\;7,\;16,\;7\right\},\;\\ \left\{13,\;10,\;13,\;10,\;13\right\},\;\\ \left\{19,\;4,\;19,\;4,\;19\right\} \end{array}\right\},\;\\ \left\{\begin{array}{c} \left\{2,\;21,\;2,\;21,\;2\right\},\;\\ \left\{8,\;15,\;8,\;15,\;8\right\},\;\\ \left\{14,\;9,\;14,\;9,\;14\right\},\;\\ \left\{20,\;3,\;20,\;3,\;20\right\} \end{array}\right\} \end{array}\right\}\left\{\begin{array}{c} \left\{\begin{array}{c} \left\{3,\;20,\;3,\;20,\;3\right\},\;\\ \;\left\{9,\;14,\;9,\;14,\;9\right\},\;\\ \left\{15,\;8,\;15,\;8,\;15\right\},\;\\ \left\{21,\;2,\;21,\;2,\;21\right\} \end{array}\right\},\;\\ \left\{\begin{array}{c} \left\{4,\;19,\;4,\;19,\;4\right\},\;\\ \;\left\{10,\;13,\;10,\;13,\;10\right\},\;\\ \;\left\{16,\;7,\;16,\;7,\;16\right\},\;\\ \left\{22,\;1,\;22,\;1,\;22\right\} \end{array}\right\},\;\\ \;\left\{\begin{array}{c} \left\{5,\;18,\;5,\;18,\;5\right\},\;\\ \left\{11,\;12,\;11,\;12,\;11\right\},\;\\ \;\left\{17,\;6,\;17,\;6,\;17\right\},\;\\ \left\{23,\;0,\;23,\;0,\;23\right\} \end{array}\right\} \end{array}\right\}$$

Furthermore, to apply the *shuffle* as an address re-order on the alternated vector reversed matrix (flipped) as a contiguous vector, Equation (28) is used, then the vector is re-indexed back to the matrix subscripts as in Equation (29):(28)$$shuffleMat\left(nFilter\right)=shuffle(flipMat_{\left(nFilter,0..\left(m-1\right)\right)},\lfloor \frac{nFilter}{2}\rfloor )$$
(29)$$TsprMat3\left[nFilter\right]=shuffelMat_{\left(Depth,Width,Height\right)}$$

Yet again for illustration, if the matrix (*TsprMat3*) is again transposed back and using the *cnt* values for a clear illustration rather than the *valSet* function response values, so as to be in comparison to previous examples in Equations (22) and (27), the matrix becomes as in Equation (30). Equation (30) shows the *shuffle* reordering of the filters when compared with the example in Equation (27), although the *valSet* value for the *cnt* value would be used in the actual implementation.
(30)$$set_{of_{cnt}}=\left\{\begin{array}{c} \left\{\begin{array}{c} \left\{0,\;23,\;0,\;23,\;0\right\},\;\\ \left\{6,\;17,\;4,\;19,\;20\right\},\;\\ \left\{12,\;11,\;6,\;17,\;4\right\},\;\\ \left\{18,\;5,\;10,\;13,\;21\right\} \end{array}\right\},\;\\ \left\{\begin{array}{c} \left\{1,\;22,\;3,\;20,\;23\right\},\;\\ \;\left\{7,\;16,\;2,\;21,\;1\right\},\;\\ \left\{13,\;10,\;9,\;14,\;19\right\},\;\\ \left\{19,\;4,\;8,\;15,\;5\right\} \end{array}\right\},\;\\ \left\{\begin{array}{c} \left\{2,\;21,\;1,\;22,\;3\right\},\;\\ \left\{8,\;15,\;5,\;18,\;22\right\},\;\\ \left\{14,\;9,\;7,16,\;2\right\},\;\\ \left\{20,\;3,\;11,\;12,\;18\right\} \end{array}\right\} \end{array}\right\}\left\{\begin{array}{c} \left\{\begin{array}{c} \left\{3,\;20,\;23,\;0,12\right\},\;\\ \;\left\{9,\;14,\;19,\;4,\;8\right\},\;\\ \left\{15,\;8,\;17,\;6,\;16\right\},\;\\ \left\{21,\;2,\;13,\;10,9\right\} \end{array}\right\},\;\\ \left\{\begin{array}{c} \left\{4,\;19,\;20,\;3,\;11\right\},\;\\ \;\left\{10,\;13,\;21,\;2,\;13\right\},\;\\ \;\left\{16,\;7,\;14,\;9,\;7\right\},\;\\ \left\{22,\;1,\;22,\;1,\;22\right\} \end{array}\right\},\;\\ \;\left\{\begin{array}{c} \left\{5,\;18,\;5,\;18,\;5\right\},\;\\ \left\{11,\;12,\;11,\;12,\;11\right\},\;\\ \;\left\{17,\;6,\;17,\;6,\;17\right\},\;\\ \left\{23,\;0,\;23,\;0,\;23\right\} \end{array}\right\} \end{array}\right\}$$

In summary, the weight initialization provides a sinusoidal bathtub distribution in convolutional layers and a uniform distribution in perceptron layers. Where the reordering provides filters, with alternating vector directions, and that reordering provides a two least neighbor arrangement. That least neighbor arrangement has stripes, spots and curves. The nominal maximum number of unique filters is as in Equation (31), although with some weight geometries of a filter the pattern repeats earlier through aliasing. This is a subject of research to extend the number of useful filters on offer.
(31)$$n_{filter}=2\cdot \left(Height_{filter}\cdot \;Width_{filter}\;\cdot \;Depth{\;}_{filter}-1\right).$$

The computational intensity of this approach is largely reordering and a *cos* function in the convolutional layers, although it is an iterative approach, as such the computation intensity increases with filter numbers and larger geometries. During prediction and back-propagation the processing intensity is unchanged; the processing is only affected during weight calculation. *Pseudocode* is included in Appendix A, for the main conv layer initialization, as a clear illustration.

## 4. Comparison of the Benchmark with the Proposed (Non-Random) Method

The presented proposed non-random initialization method achieves 93.28% in a single epoch with no shuffle, +3.705% better than the existing random method. 93.77% accuracy is achieved when a shuffle is used in a single epoch, again using the proposed non-random method, which is a 2.642% gain over the benchmark of the existing random method. Then 97.5% (+0.599% over the existing random method) when five epochs are used. Those results are within Table 2 shown in bold.

From Table 2 the best gains are achieved in the first epoch, which is the epoch that occurs after the weight initialization. Less relative gain is achieved in further epochs, as the learning is occurring longer after the initialization in the subsequent epochs, diminishing its influence but inheriting the earlier learning. An interpretation is the subsequent learning is more equivalent but earlier learning has a higher benefit, as it may be using more of the dataset more effectively in the first epoch.

### 4.1. Comparison of the Weights before and after Learning

In Figure 4 (left), is the existing random method benchmarks weights before learning of the first convolutional layer, and in Figure 4 (right), are the same filter weights but after learning.

In Figure 4, the existing random initialization method form has a speckled appearance in each filter, and it may also be noticed that there are high similarities in the filters, between the before and after learning. Suggesting that the initial condition has a dominant effect in the subsequent learning, even over a large dataset. This is perhaps why the variation in accuracy with different random sequences is visible over regularization. As the initial speckled positions in the filter relate to positions in image features, affecting the resultant performance of that filter from the outset. Meaning that the filter organization is important to the performance, rather than just statistical equivalence. Furthermore, looking carefully, adaption can be noticed between the before in Figure 4 (left) and after learning in Figure 4 (right). Illustrated by when the initial weights are subtracted from the learned weights, the adaption can be seen more clearly in Figure 5.

Consistent with Figure 5, convolutional filters examined after learning may be expected to have stripes, spots and perhaps curves, that may be used in edge detection of feature extraction, that will be hierarchically connected and organized to form shapes in later layers. Combined with this, striped and curved patterns may be more conducive in shape detection, that vary from filter to filter and be a less specific specialized mapping in larger filter geometries. In filter generation stripes and curves from a modulation may orientate directions with different selections of width bounding values as they wrap by the maximum width value of the filter, too. Considering the proposed method, in Figure 6 (left) is the proposed non-random method weight filter initialization, again produced for the first convolutional layer. In Figure 6 (right) is those same weights of the filters but after learning has been conducted, as the update to those filters.

It may be noted that there are also some similarities between the before in Figure 6 (left) and after learning in Figure 6 (right) with the proposed method, reinforcing the dominance of the initial condition assertion. But those similarities are less, and this method is higher performing than the existing random method. When subtracted to expose the learned adaption in Figure 7, there are similarities with Figure 5. The subtraction of the before learning initialization from the learned weights might be thought of as what has been learned but is actually more formally what has been adapted from the initial condition in nudges of values in optimization iterations.

Thus it may also be noticed that there are also some similarities in what has been relatively adapted, between the random and non-random methods in Figure 5 and Figure 7. In fact, some equivalent relative filter adaptations between them can be noticed. So it follows that the initial condition, therefore, has an effect originating from their arrangement from the outset of learning, and different arrangements will affect the after learning result. However, the relative adaptation from the methods (both random and non-random) has some equivalence indicating that it is a similar design implementation arrived at from the same dataset, rather than a different implementation. This is reassuring as the dataset, model architecture and algorithms are unchanged, and it is the initialization alone that has been modified, and that is what is responsible for the increased accuracy.

### 4.2. Comparison of the Optimization Objective (Loss) in Learning

When the losses are compared during learning, of the first epoch, as the epoch after initialization, where the initial learning occurs. Then the loss does reduce quicker with the proposed non-random method, owing to the stripes and curves in the initial condition, being pre-disposed to the application of image categorization. Note that the loss is shown as it is the optimization objective. See Figure 8 (left) for the existing random method, and Figure 8 (right) for the proposed non-random method when shuffled.

Figure 9 also makes the comparison with the existing random method in Figure 9 (left), and the proposed non-random method in Figure 9 (right). In comparison to the dataset shuffling in Figure 8, and the un-shuffled dataset results in Figure 9, the proposed non-random method has achieved the lower loss quicker in learning in both cases, and is noted at the batch 100 point. Thus earlier learning has benefited relatively in the proposed non-random method, using more of the dataset more effectively from the outset of learning, regardless of the dataset shuffling.

### 4.3. Comparison to the State-of-the-Art (He et al. Initialization)

There is however an enquiring question raised, and that is: although it is a departure from the benchmark model, would the proposed method be robust to He et al. [27] initialization limits instead. As He et al. initialization is regarded as the state-of-the-art. The uniform He et al. initialization limit values are calculated in Equations (32)–(34):(32)$$ConvLayer1=\sqrt{\frac{6}{\left(5\cdot 5\cdot 1\right)}}=0.48989795,$$
(33)$$ConvLayer2=\sqrt{\frac{6}{\left(5\cdot 5\cdot 32\right)}}=0.08660254\;\mathrm{and}$$
(34)$$DenseLayer=\sqrt{\frac{6}{\left(4\cdot 4\cdot 64\right)}}=0.07654655.$$

As before with the Glorot/Xavier limits the He et al. limit values are rounded to eight decimal places [32], so as to be compatible in comparison. The cross-validation results using the He et al. initialization limits and the proposed method are presented in Table 3 in columns: ‘Loss’ and ‘Accuracy’. The relative gain percentage of the proposed non-random method, between He and Glorot/Xavier limits is also within the column ‘Glorot (Non-Rand) [Table 2]’, as relative to Table 2, as the gain of He initialization with the proposed method. Table 3, also for completeness shows the comparison gain using the proposed non-random method, and the existing random method, both with He et al. initialization, which is in the column ‘He (Rnd)’, as the gain of the proposed method with He initialization.

In all cases in Table 3, the proposed method offers a positive accuracy gain advantage in cross-validation. Again, the greatest increases in accuracies are in both the first epoch cases shown in bold. The proposed method has an advantage in cross-validation accuracy when either applied to Glorot/Xavier or the current state-of-the-art of He et al. limit values and is repeatable and deterministic. Again that is an additional advantage for the development of dependable safety critical applications, and also is an advantage within smart sensors using image classification.

## 5. Fast Sign Gradient Method (FSGM) and Transferred Learning Approach

This section will further examine the proposed non-random and existing random methods, using a transferred learning and FSGM approach. Then demonstrate applicability to color images in a dissimilar architecture. FSGM with transferred learning is a convenient approach to control distortions in a transferred learning dataset from the FSGM’s epsilon (*ε*) value. The FSGM model defense with a transferred learning approach is used rather than other approaches, as it will demonstrate the influence of how compromised the transferred learning is, over the updated further learning, and the transferred learning approach will provide an indicator of those influences with a varying controlled magnitude of error distortion via the FSGM’s epsilon (*ε*) value.

### 5.1. The FSGM Transferred Learning Approach

A modern theme in neural networks is the area of perturbation attacks using the Fast Sign Gradient Method (FSGM) attack proposed by Ian Goodfellow [34,35]. This is an attack that can cause miss-classifications with an effect that may not be humanly perceivable. The attack itself can have a strength value of the attack controlled by an error magnitude value epsilon (*ε*) [34,35] as in Equation (35):(35)$$x^{\prime }=x+\varepsilon \cdot sign\;\left(\nabla_{x}J\left(\theta ,x,y\right)\right)$$

A modification to the FSGM attack equation is made to avoid out of scale numbers in the image after perturbation, by clipping the perturbation image pixel values between 0 and 1, as in Equation (36). This is to be compatible in comparison to the non-perturbed image pixel scales, that were also scaled between values 0 and 1 in the Torres model [29]:(36)$$x^{\prime }=max\left(\min\left(x+\varepsilon \cdot sign\;\left(\nabla_{x}J\left(\theta ,x,y\right)\right),1.0\right),0.0\right)$$
when the epsilon (*ε*) value is small the attack can be a deception (or *spoofing*) coursing miss-classification to another number assignment other than a human would. Also, when the epsilon (*ε*) value is large it can cause a *denial of service* (*DoS*) to the human while still having a classification in the computer. It might be noted, that such approaches might also have applications to encryption and hidden messages.

The FSGM transferred learning approach used is by Theiler [36], and it explains the attack with examples, and importantly uses the same MNIST dataset. See Figure 10 for the Theiler and Torres architecture as integrated, with three experiment test points used in this paper’s analysis.

The Theiler attack dataset sizes [36] are thus used as the MNIST dataset is the same, although is adapted to the Torres benchmark’s number of epochs [29] instead. The epochs used in transferred learning are set to be half that of the Torres baseline benchmark case, instead of half the Theiler number of epochs, as is the case from Theiler’s initial learning model. Such that the amount of back propagation is relatively similar to the transferred learning from Theiler’s model but applied to the Torres model.

The relevance of the FSGM attack to this paper is a hypothesis that FSGM attacks could be effective partially from the less humanly perceivable noise content in an image dataset rather than the useful information in the dataset alone. Furthermore, that that noise could be noise re-colorized by the random numbers as a noise source in the initialization state when the weights are multiplied with the activations. It follows that if an initialization state has less random content that could be thought of as noise, then the subsequent learning may have less opportunity for unintentional noise re-colorization as a result. Thus the effect could affect the compromise of the original training after defense if the adversarial attack dataset is used to retrain the model to protect it. The FSGM approach also provides a controlled distortion via the epsilon (*ε*) value.

In Figure 11 (left and right) are the first 20 images of the FSGM perturbation attack datasets. They are generated from the sign of the gradient of the loss of a true prediction and an image, In Figure 11 (left) is the generation from the existing random form, and in Figure 11 (right) from the proposed non-random method instead. Along the column’s axis are the first 20 images of each dataset, and in each row, as perturbed with images with an increasing epsilon value of 0.0 to 1.0 in steps of 0.05. In green are the perturbation images that are correctly classified against their original tag, and in red are the perturbation images that are miss-classified.

Figure 11 shows that the low perturbation epsilon error values images (at the first rows) are humanly recognizable and the high epsilon error values images (at the last rows) are not humanly recognizable. At the point that the miss-classifications occur, the strength of the competing image’s pixels becomes more apparent, and traditional de-noising and image value clipping would enhance the image classification in many cases of both methods. The subject of noise and noise injection, but as a method for defense from FSGM is proposed in 2020 by Schwinn et al. [37], but their approach requires a learning regularization step that couples it to the dataset. Although the Schwinn et al. [37] approach is of interest, in a non-dataset coupled form, and perhaps as an augmentation to the epoch scheme. However, this paper’s approach is purely to examine the transferred learning compromise with a reduction in noise re-colorization opportunities. Thus applying the same perturbed FSGM attack method with different epsilon error strengths, a comparison in accuracy and loss between the existing (random) and proposed (non-random) methods are presented in Figure 12, Figure 13 and Figure 14.

### 5.2. The Undefended Model, Attacked by FSGM

Figure 12 (left) and Figure 12 (right) are results (at the experiment test point 1) in accuracy and loss, from cross-validation with a generated validation adversarial attack perturbation dataset of 10,000 images using the FSGM approach, and applied to an undefended model. In Figure 12, both the existing (random shown in red) and proposed (non-random shown in blue) methods are using the higher performing He et al. initialization limit values, and with controlled steps of the epsilon (*ε*) distortion value. In Figure 12 (left) in both initialization cases, the results are almost identical in accuracy, with an undefended model, signaling a similar generalization was achieved under attack. Both the existing (random) and proposed (non-random) methods are susceptible to FSGM attack, and in both cases, the susceptibility is greater as the epsilon value gets larger, signaled by the cross-validation accuracy lowering.

In Figure 12 (right) ideally, the losses may be low showing regularization is still effective, i.e., it reduces the variance, despite the perturbation attack. However, in this attack the sign of the gradient is taken from a calculated loss of a true prediction and an input image, that gradient is then applied to an image as a perturbation image, therefore the loss rises as the epsilon value is raised, as the controlled distortion. The proposed (non-random) method has a lower loss than the existing (random) method, at larger epsilon values. Suggesting that the perturbation of features in the attacks is less sensitive to regularization in the proposed method, although this is at epsilon values where the generalization (accuracy) has diminished in the attack.

### 5.3. Defending a Model from FSGM

The Theiler [36] approach to defending a model from an attack is to transfer learning and include adversarial perturbation attack examples into a further training dataset and generate a further attack validation dataset. Then cross-validate with both validation datasets (i.e., both attack and non-attack datasets). This makes a comparison in accuracy and loss after the model defense. As such the original model is further trained with 20,000 training adversarial perturbation attack images to defend it over two epochs, examined with controlled increments of the epsilon (*ε*) value. In the experiment approach, the epsilon (*ε*) value is stepped, providing a controlled distortion in the transferred learning. Such that the controlled distortions can be examined after transferred learning, and reveal how compromised both methods are after the learning transfer.

### 5.4. Examining the Transferred Learning Adaption

The results from the existing (random) and proposed (non-random) methods are shown for experiment test point 2. In Figure 13 the performance against the validation attack perturbation dataset is shown. Such that the transferred learning of the original dataset is transferred to an attack dataset to evaluate the performance. In Figure 13, are shown the accuracy and losses of the defended model cross-validated with the validation attack dataset, and shows the existing (random) and proposed (non-random) methods, have similar performance with low epsilon values. Figure 13 (left) shows that with larger epsilon values, the cross-validation accuracy of the attack dataset is higher with the existing method, showing the existing method prefers the further generalization in retraining after model defense at higher distortions (less numerically similar). In Figure 13 (right) the losses are also lower in the existing method indicating that they are slightly more regularized in the existing method with less numerically similar datasets. Both loss and accuracy indicate that the new attack dataset is being embraced more in the existing method when the new attack dataset is less numerically similar.

### 5.5. Examining the Model Compromise in Defense to the Original Cross-Validation Dataset

Figure 14 shows the results after model defense with the adversarial FSGM perturbation attack dataset, but cross-validated with the original non-attack dataset (at experiment test point 3). This is to show how compromised the models are to the original non-attack dataset after model defense via transferred learning.

Figure 14 shows the defended model with the original dataset cross-validation in accuracy and loss. It also shows that at greater epsilon error magnitudes the difference between the existing (random) and proposed (non-random) methods are divergent. The proposed (non-random) method being the higher accuracy in Figure 14 (left) and lower loss in Figure 14 (right). Meaning that the proposed (non-random) method has suffered less when re-trained with the perturbed dataset to defend it, i.e., the transferred learning has less affected the accuracy to the original dataset, most notably with larger distortions (i.e., a numerically dissimilar dataset). Thus the proposed method has retained more of the original learning from the dataset that was transferred.

### 5.6. FSGM Attacks with Varying Epsilon Values

In Figure 12, Figure 13 and Figure 14, each FSGM perturbation dataset used a constant epsilon value that was progressively increased in datasets. In Table 4 are the results from both the existing (random) and proposed (non-random) methods, but with a randomly varying epsilon value in both the cross-validation and training FSGM attack datasets. As a mixture of numerical dissimilarity in a single dataset, used in transferred learning.

In Table 4 the cross-validation accuracy with the proposed (non-random) method is greater with a lower loss when cross-validated with the original non-attack validation images. This is shown in bold, within the column: ‘Non-Attack Cross-Validation Dataset’. However, the cross-validation accuracy is higher with a lower loss with the existing method using the attack cross-validation dataset. This is also shown in bold, within the column: ‘Attack Cross-Validation Dataset’. Table 4 demonstrates that the proposed non-random method has been less compromised to the original learning that was transferred. Also that the existing method, by contrast, had higher accuracy and lower loss with the retraining attack dataset. Thus may have embraced the subsequent attack dataset more, but has been more compromised to the original training to a greater extent. These findings were also shown in Figure 12, Figure 13 and Figure 14, and those results also showed a greater difference with constant epsilon values, and the most extreme difference was with the larger epsilon (*ε*) values, as numerically dissimilar datasets.

### 5.7. Color Images and Dissimilar Model Architectures

Demonstrated with a color image dataset originating from the Multi-Type Aircraft of Remote Sensing Images (MTARSI) dataset [38], of aircraft on runways and taxi-ways. The original MTARSI dataset [38] was modified into 42 classifications, and extra data augmentation in those classifications, which has been made available as MTARSI 2 [39]. It is an example of an unbalanced dataset, with challenges of different light and viewing angles. This dataset and model also highlight the different dense layer initialization algorithm selections required, based on the layer prior in a dissimilar architecture.

Using the state-of-the-art He et al. initialization limit values, the model in Figure 15 provided 85.57% cross-validation accuracy with the proposed (non-random) method. This was higher than 82.28% provided with random numbers. Also in the first epoch after initialization, the proposed non-random method achieved 54.69%, which was also higher than the 44.89% with a random number initialization.

## 6. Discussion and Conclusions

This work focused on repeatable determinism to support mission and safety critical systems that use convolutional networks with mixes of both perceptron and convolutional layers.

### 6.1. Proposed Method in Neural Networks

The proposed method is applicable to deep convolutional networks for repeatability, however also achieves a higher accuracy in a single learning session, with a computational initialization state number sequence that has been designed to be more conducive to image classification. The proposed method has a finite number sequence set that is not coupled to the dataset via sampling, which may be closer to a general case. The losses during learning show a quicker reduction using the proposed form, and result in a higher accuracy. The repeatable deterministic property also provides no variation in learning sessions, aiding the speed of development of a model. The proposed method is complementary to existing methods, replacing only the random numbers in those methods. The proposed non-random method provided a higher cross-validation accuracy against the existing random number method, and when used with Glorot/Xavier limits of the benchmark models achieved an extra 3.705% un-shuffled, and 2.642% shuffled in the first epoch. Thus dataset order is still a significant effect, but the proposed method was tolerant. The proposed method is also robust to He et al. initialization value limits as the current state-of-the-art, and when used with the proposed method offered an accuracy gain of 5.13% shuffled and 4.27% un-shuffled in the first epoch. The proposed method was also demonstrated in a dissimilar architecture with color images and still provided advantages in the first epoch of ~10%. A subject of research is increasing the number of filters on offer, and making them tolerant to the hyper-parameters in more architectures. Such that the proposed method is adapted to an architecture generally and is not constrained to it, providing the best filters for any model architecture.

### 6.2. Proposed Method in FSGM with Transferred Learning

When applied to FSGM with transferred learning the proposed (non-random) method has the advantage of being less compromised, noted with larger epsilon values. In this context, larger epsilon values may equate to numerically dissimilar datasets. The results do show that the proposed method when transferred in this order, provided a lower compromise to the original programming with a numerically dissimilar dataset, and achieved an accuracy of ~31%, compared with ~9% of the existing method. The bias to original training is favored, illustrated by, in the Torres baseline if the transferred learning was to extend the training to a letter set from A to F as the hexadecimal set. Then the existing random method may be biased to replacing the training of numbers with letters. Whereas the proposed non-random method may be more biased to extending the training to letters, while still retaining more of what was learned from numbers.

## Figures and Tables

**Figure 1 sensors-21-04772-f001:**
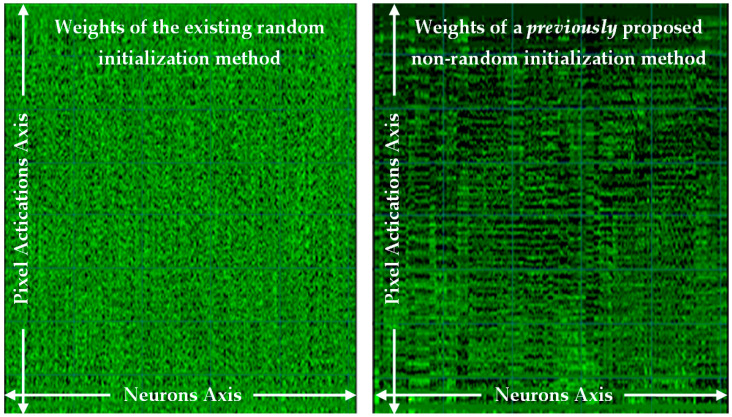
Weight matrix after learning, results from a perceptron only network, (**left**) is an existing random number method, and (**right**) a non-random method reproduced from previous work [16].

**Figure 2 sensors-21-04772-f002:**
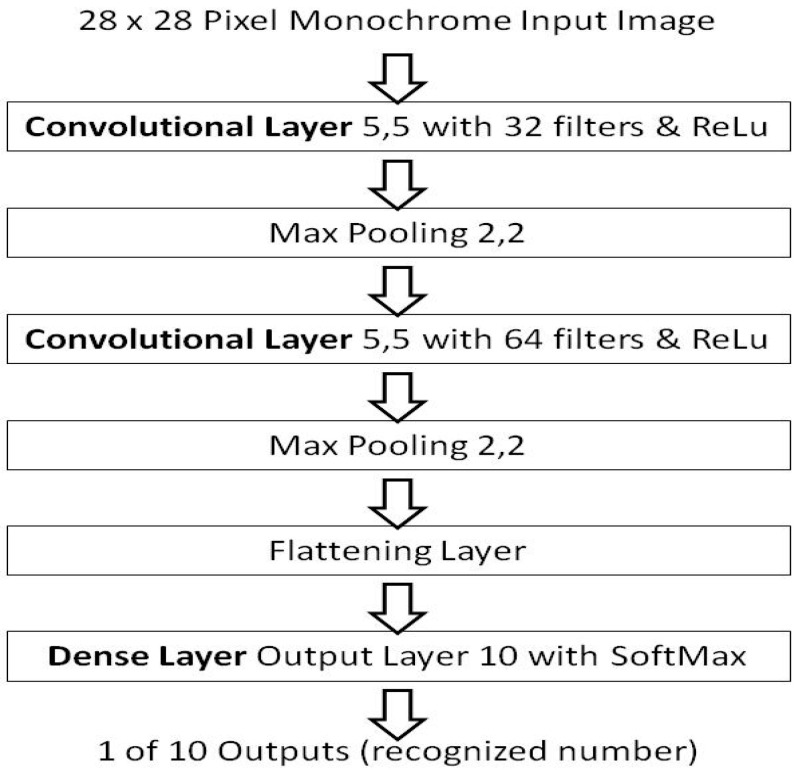
The model architecture of the benchmark model, by Torres.

**Figure 3 sensors-21-04772-f003:**
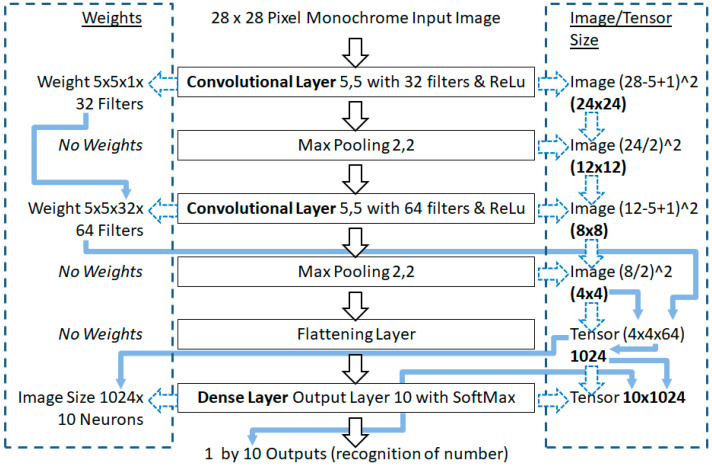
Weight and image size adjustments of layer parameters in the Torres benchmark model.

**Figure 4 sensors-21-04772-f004:**
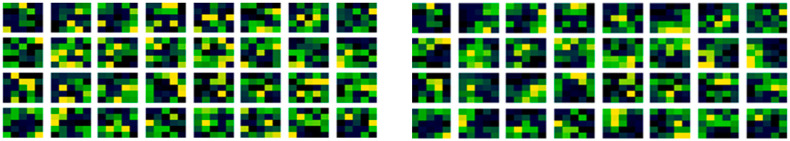
Initial and learned weights in the first conv layer with the existing random method.

**Figure 5 sensors-21-04772-f005:**
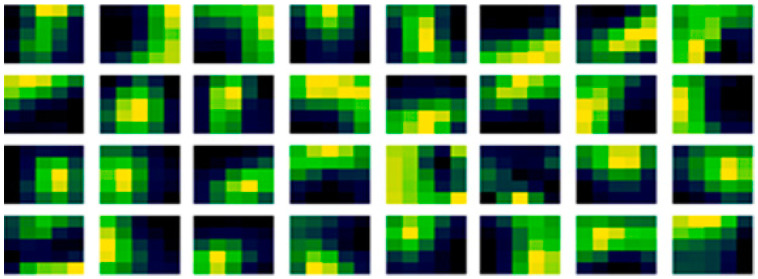
Learned filter weight adaption-updates of the existing random method.

**Figure 6 sensors-21-04772-f006:**
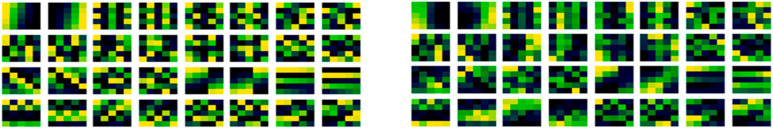
Initial and learned weights in the first conv layer with the proposed non-random method.

**Figure 7 sensors-21-04772-f007:**
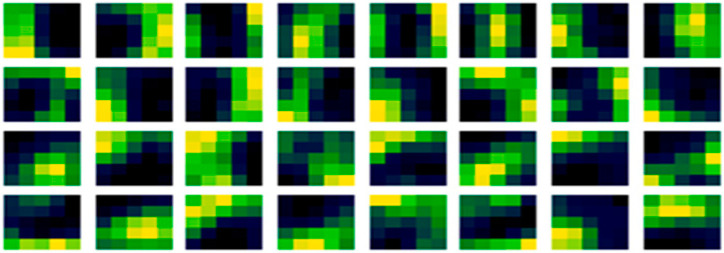
Learned filter weight-adaption updates with the proposed method.

**Figure 8 sensors-21-04772-f008:**
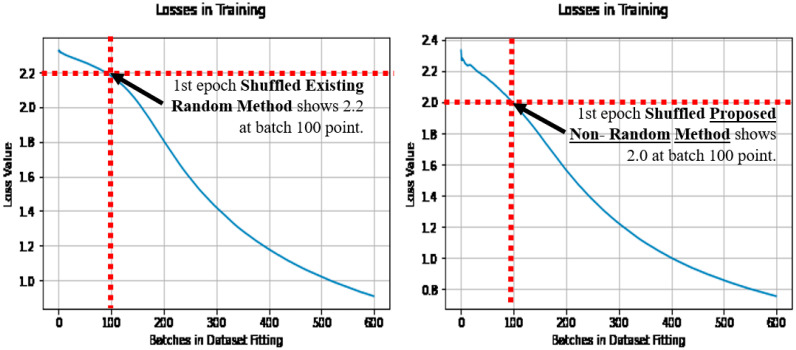
Losses over batches in fitting, when shuffled (with the existing method (**left**) and the proposed method (**right**)).

**Figure 9 sensors-21-04772-f009:**
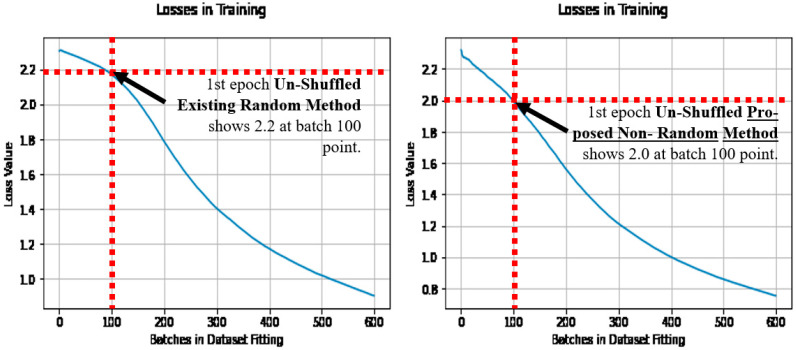
Losses over batches in fitting, when un-shuffled (with the existing method (**left**), and the proposed method (**right**)).

**Figure 10 sensors-21-04772-f010:**
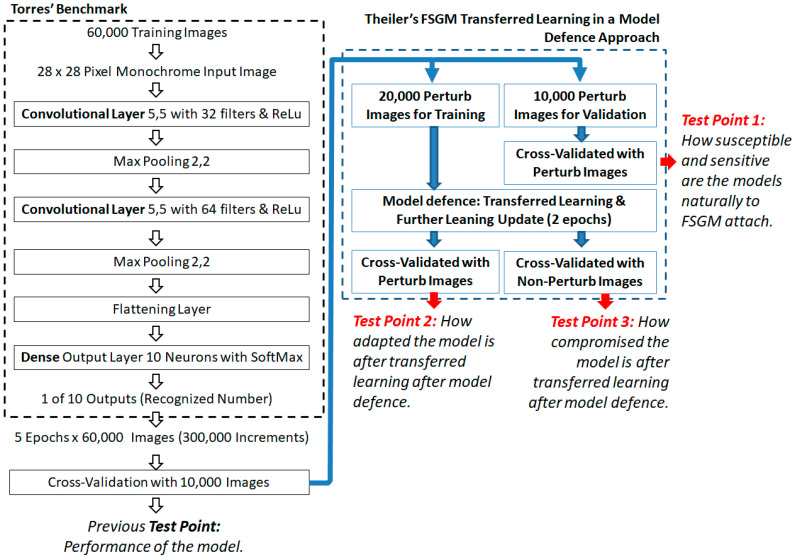
Theiler’s FSGM transferred learning experiment model [36], as added to the Torres benchmark model showing the experiment test points.

**Figure 11 sensors-21-04772-f011:**
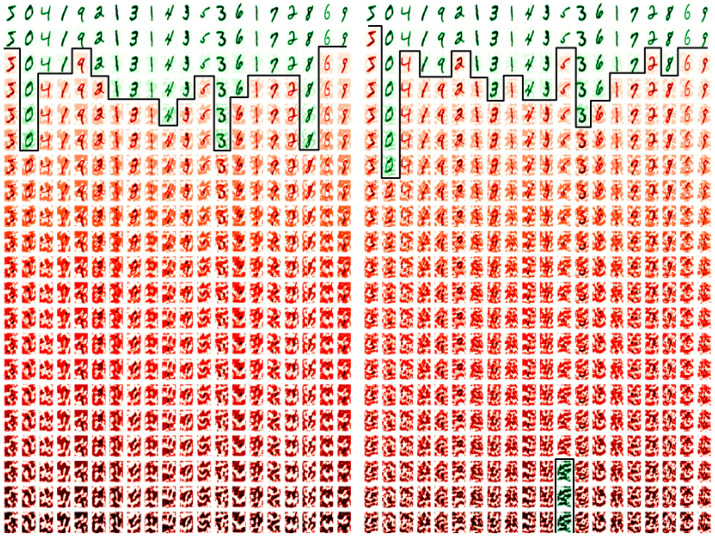
First 20 images of the FSGM attack image examples: using the existing method (**left**) and the proposed method (**right**).

**Figure 12 sensors-21-04772-f012:**
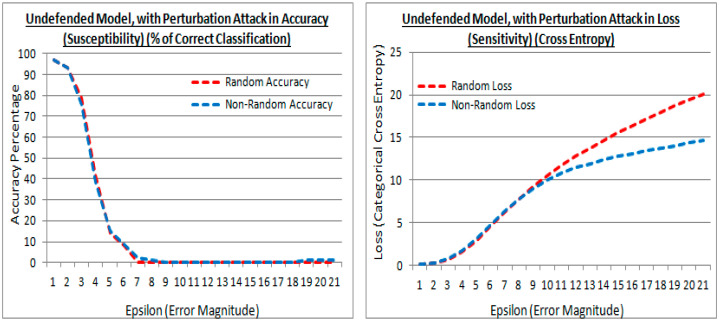
Accuracy and losses of an undefended model with epsilon increments under attack, for both the existing (random) and proposed (non-random) methods.

**Figure 13 sensors-21-04772-f013:**
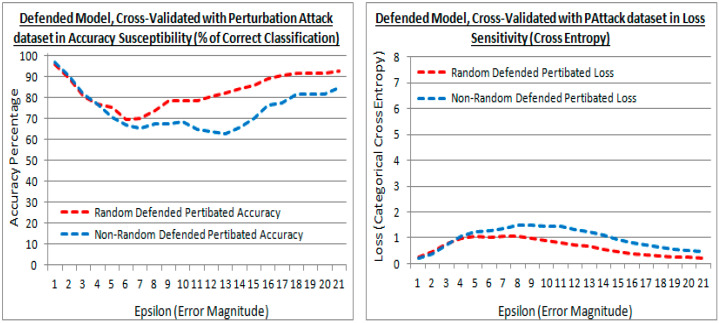
Accuracy and losses of a defended model with epsilon increments against the validation attack dataset, for both the existing (random) and proposed (non-random) methods.

**Figure 14 sensors-21-04772-f014:**
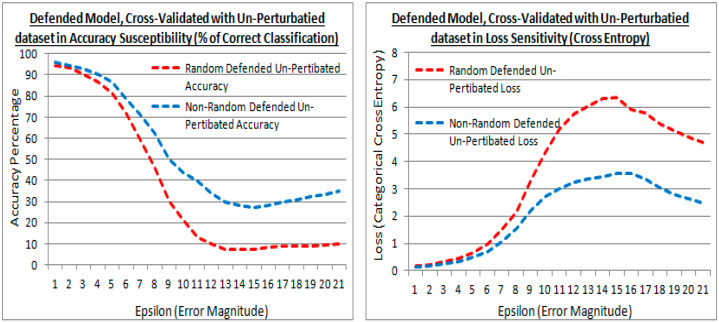
Accuracy and losses of a defended model with epsilon increments against the original non-attack cross-validation dataset, for both the existing (random) and proposed (non-random) methods.

**Figure 15 sensors-21-04772-f015:**
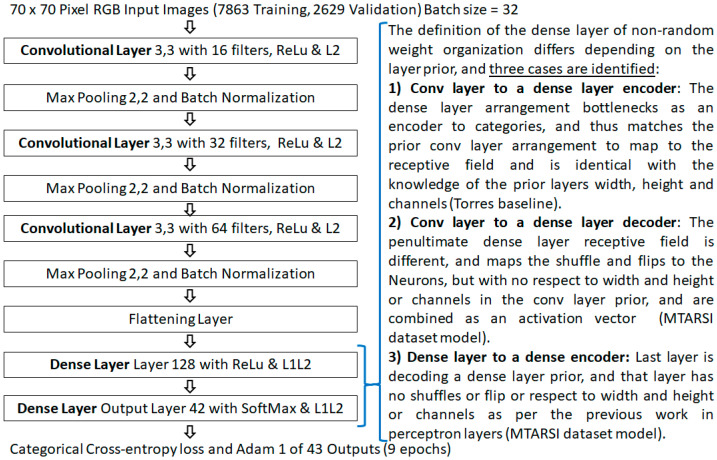
Color image model with dissimilar architecture, used for the MTARSI 2 [39] dataset.

**Table 1 sensors-21-04772-t001:** Weights and image size parameters in each layer of the benchmark model by Torres.

Layer	Filter/Pool/Neurons	Depth	Image/Tensor Size	Weights
Input 28 × 28 × 1	N/A	1 (B/W image)	28 × 28 (748)	N/A
Conv Layer 1	5 by 5 by 32 filters	1	24 × 24 (576)	800
Max Pooling	2 by 2	32	12 × 12 (144)	N/A
Conv Layer 2	5 by 5 by 64 filters	32	8 × 8 (64)	51200
Max Pooling	2 by 2	64	4 × 4 (16)	N/A
Flatten Layer	N/A	1	1 × (4 × 4 × 64) 1024	N/A
Dense Layer	10	1	10 × 1024 (10,240)	10240

**Table 2 sensors-21-04772-t002:** Gains of the proposed non-random initialization method over the existing benchmark.

Epochs	Accuracy (Cross-val.)	Loss (Cross-val.)	Gains over Existing (Random) Method
**5 Shuffled**	**97.5%**	**0.085728347**	**+0.599%** (Cross-validation gain)
4 Shuffled	97.11%	0.097854339	N/A
3 Shuffled	96.85%	0.114757389	N/A
2 Shuffled	95.96%	0.141269892	N/A
**1 Shuffled**	**93.77%**	**0.230065033**	**+2.642%** (Cross-validation gain)
**1 No Shuffle**	**93.28%**	**0.230725348**	**+3.705%** (Cross-validation gain)

**Table 3 sensors-21-04772-t003:** He et al. limits with the proposed initialization method and gains in comparison.

Epochs	He et al. (Non-Rnd) Measure		He with Proposed Method Gains Over:	
	Loss	Accuracy	Glorot (Non-Rand) [Table 2]	He (Rnd)
5 Shuffled	0.082669578	97.55%	+0.05%	+0.7%
4 Shuffled	0.093996972	97.19%	+0.08%	+0.91%
3 Shuffled	0.10997723	96.97%	+0.12%	+1.49%
2 Shuffled	0.134461805	96.15%	+0.19%	+1.83%
1 Shuffled	0.214723364	94.11%	**+0.34%**	**+5.13%**
1 No Shuffle	0.217569217	93.57%	**+0.29%**	**+4.27%**

**Table 4 sensors-21-04772-t004:** Results from a single dataset that included random epsilon values, for both the existing (random) and proposed (non-random) methods.

Initialization Method Used Prior to Model Defense via Transferred Learning	Non-Attack Cross-Validation Dataset		Attack Cross-Validation Dataset	
	Loss (Cross-Val)	Accuracy (Cross-Val)	Loss (Cross-Val)	Accuracy (Cross-Val)
Proposed (Non-Random) Method	**0.9854**	**67.01%**	1.3331	61.82%
Existing (Random) Method	1.2736	61.05%	**0.8366**	**78.41%**

## Data Availability

See reference [39] for updated MTARSI 2 dataset.

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
