# Peer review of "Deep ConvNet: Non-Random Weight Initialization for Repeatable Determinism, Examined with FSGM†"

_sensors, 2021, doi:10.3390/s21144772_

Round 1
Reviewer 1 Report
This article describes a non-random weight initialization method which is repeatable and deterministic.
The approach seems to be novel and superior over existing methods. In particular, it is applicable within convolutional layers directly and it gives accurate results in a single epoch.
However, Section 2 describing the related work is too short to understand the related research flow.
Supplementing details of the described recent works and adding several more works will improve the article.
Author Response
Thank you indeed for your review, it really helped me.

Reviewer 2 Report
In the reviewed paper, the authors show deep convnet which is based on non-random weights. In general, the paper in its current state is not ready for publication. In my opinion, the novelty is very weak, the same issue is with the introduction/related works and experimental sections. I think the authors should extend the proposition to show the novelty. Some of my comments:
1) The abstract is very long and shows no novelty, no result information.
2) The introduction and used bibliography should be extended to show also other application of CNN-based networks and their applications in a real environment like in "Automatic ship classification for a riverside monitoring system...". Moreover, analyze the latest solutions in LSTM, and the latest weights selection for neural networks.
3) Sec. VIII needs more theoretical analysis of your proposition like time/computational complexities.
4) There should be pseudocodes for a better understanding of the implementation process.
5) Is your proposition valid for all types of CNN models?
6) The charts and figures should be improved in terms of quality.
7) The experimental section needs to be analyzed on more different architectures.
8) Compare your proposition with state-of-art to show the advantages of your proposal. Use other research from the last 3-4 years with the selection of weights in neural models.
9) Compare your solution with other models like learning transfer, etc., and show the convergence curves.
10) Use another database in the experimental section. There must be some comparison and analysis of color images.
11) Show the feature maps of color images obtained by your proposition and other known models like VGG16, etc. Analyze the results and discuss them.
Author Response

(The authors gave the same response as above.)

Reviewer 3 Report
This paper proposed an interesting work on initializing deep convolutional network model with non-random sequence
to accommodate fast learning rate, particularly at the beginning phase. In addition, the proposed method is tested against FSGM attack.
However, the generalization of the proposed non randomized initialization method shall be evaluated on other datasets, in addition to MNIST,
and with different configurations, i.e., image sizes, # of layers, # of nodes in each layers. Without these evaluations, the merit of the proposed method could not be fully assessed.
In addition, there are several concerns regarding the current version of the manuscript. First, the organization of the paper shall be improved. It is not until section VIII that the proposed non-random weight sequence described. Secondly,the paper is too lengthy with much content (p. 1-9) that are related work.
Also, Table 1 is very little relevant to the presented work. It shoud not be included in the main paper
(can be placed in appendix if neccesary). In table 4, the loss and accuracy of the proposed method is not shown -- only the gain in comparison with [17]
is depicted.
Author Response

(The authors gave the same response as above.)

Reviewer 4 Report
In this paper, authors presented a non-random weight initialization method in convolutional layers of neural networks examined with the Fast Gradient Sign Method (FSGM) attack. However, there are some limitations that must be addressed as follows.
- The abstract is not attractive. Some sentences in abstract should be modified to make it more attractive for readers. Authors have started every sentence from ‘this paper’ or ‘the proposed method’. Authors should just discuss the main novelty. Also remove extra spaces.
- In Introduction section, it is difficult to understand the novelty of the presented research work. This section should be modified carefully. In addition, the main contribution should be presented in the form of bullets. Also remove the subtitle ‘A structure of the paper’ and summarize this section.
- Authors have mixed everything. The contribution section should be the part of introduction Section.
- Related work should be extended by including more existing work (Traffic accident detection and condition analysis based on social networking data’, ‘An intelligent healthcare monitoring framework using wearable sensors and social networking data’, ‘ABCDM: An Attention-based Bidirectional CNN-RNN Deep Model for sentiment analysis’,’BiERU: Bidirectional Emotional Recurrent Unit for Conversational Sentiment Analysis’
- Figure 3 is difficult to understand. It should be more deeply explained.
- The same font size and style should be used for equations.
- Captions of the Figures not self-explanatory. The caption of figures should be self-explanatory, and clearly explaining the figure. Extend the description of the mentioned figures to make them self-explanatory.
- The whole manuscript should be thoroughly revised in order to improve its English.
Author Response

(The authors gave the same response as above.)

Round 2
Reviewer 2 Report
All my comments have been addressed. The paper is very good and can be accepted in its current form.
Reviewer 3 Report
The authors have addressed the comments carefully.
Reviewer 4 Report
The authors have addressed my all comments. I have no further comments. Therefore, this paper can be accepted in its present form